# Determination of 77 Multiclass Pesticides and Their Metabolitesin *Capsicum* and Tomato Using GC-MS/MS and LC-MS/MS

**DOI:** 10.3390/molecules26071837

**Published:** 2021-03-25

**Authors:** Harischandra Naik Rathod, Bheemanna Mallappa, Pallavi Malenahalli Sidramappa, Chandra Sekhara Reddy Vennapusa, Pavankumar Kamin, Udaykumar Revanasiddappa Nidoni, Bheemsain Rao Kishan Rao Desai, Saroja Narsing Rao, Paramasivam Mariappan

**Affiliations:** 1Pesticide Residue and Food Quality Analysis Laboratory, University of Agricultural Sciences, Raichur 584 104, Karnataka, India; bheemuent@rediffmail.com (B.M.); pallavipath@gmail.com (P.M.S.); csreddy.vennapusa@gmail.com (C.S.R.V.); pannu49500@gmail.com (P.K.); udaykumarnidoni@yahoo.co.in (U.R.N.); dr@uasraichur.edu.in (B.R.K.R.D.); saroja95@yahoo.com (S.N.R.); 2Pesticide Toxicology Laboratory, Tamil Nadu Agricultural University, Coimbatore 641 003, Tamil Nadu, India

**Keywords:** multiclass pesticides, *Capsicum*, tomato, QuEChERS, LC-MS/MS, GC-MS/MS

## Abstract

A quick, sensitive, and reproducible analytical method for the determination of 77 multiclass pesticides and their metabolites in *Capsicum* and tomato by gas and liquid chromatography tandem mass spectrometry was standardized and validated. The limit of detection of 0.19 to 10.91 and limit of quantification of 0.63 to 36.34 µg·kg^−1^ for *Capsicum* and 0.10 to 9.55 µg·kg^−1^ (LOD) and 0.35 to 33.43 µg·kg^−1^ (LOQ) for tomato. The method involves extraction of sample with acetonitrile, purification by dispersive solid phase extraction using primary secondary amine and graphitized carbon black. The recoveries of all pesticides were in the range of 75 to 110% with a relative standard deviation of less than 20%. Similarly, the method precision was evaluated interms of repeatability (RSDr) and reproducibility (RSD_w_R) by spiking of mixed pesticides standards at 100 µg·kg^−1^ recorded anRSD of less than 20%. The matrix effect was acceptable and no significant variation was observed in both the matrices except for few pesticides. The estimated measurement uncertainty found acceptable for all the pesticides. This method found suitable for analysis of vegetable samples drawn from market and farm gates.

## 1. Introduction

Vegetables are major components of the human diet which provide the vitamins (A, B_1_, B_6_, B_9_, C and E), carbohydrates, proteins, antioxidants (sulphoraphane, nasunin, allicin and diosgenin) and minerals needed for a balanced diet [1]. As a part of healthy diet vegetables reduce the risk for heart diseases, obesity, type 2 diabetes, lower the blood pressure, reduce the blood cholesterol levels and reduce the development of kidney stones [2,3]. Vegetables are grown using different groups of pesticides because of the damage caused by insect pests and diseases from the seedling stage and up to the fruiting. For achieving higher production pesticides sprayed either singly or in combination are taken-up and may contribute to the generation of pesticide residues. The concentration of pesticide residues may have impacts on the environment [4]. Pesticide residues in many vegetables exceeding the maximum residual limits (MRLs) may cause health hazards. Consumption of fresh vegetables is the primary route of pesticide intake through the diet [5,6]. Several reports have confirmed the contamination of vegetables with residues of pesticides such as neonicotinoids, synthetic pyrethroids, organophosphates and carbamates [7].

Tomato (*Lycopersicon esculentum* Mill.) is the most common vegetable crop cultivated in tropical and subtropical regions for fresh market and processing purposes. It is an essential vegetable of the world diet and used in different forms viz., raw, home-cooked, and as canned products, juices or pastes [8]. Due to its tenderness and softness, it is more vulnerable to insect pests and diseases compared to other crops. Insect pests causing significant damage to tomato are fruit borers, whiteflies, jassids, thrips and serpentine leaf minerswhich are responsible for fruit yield losses ranging from 14 to 45% [9]. Protection of tomatoes is commonly carried out by scheduled spraying using different kinds of pesticides such as systemic, contact or both. Several reports on the occurrence of pesticide residues and their derivatives in different vegetables including tomatoes are available from previous studies [10,11].

Since ancient times peppers (*Capsicum annuum* L.), are another important vegetable used as a natural food color, rich in phyto-nutrients, vitamins (A, C and E), flavonoids and capsaicinoids having potential health benefits [12,13,14]. Apart from growing them in the open field, nowadays their production is highly restricted to greenhouse. The abiotic conditionsprevailing in the open field and greenhouses are responsible for diseases, insects and nematode injuries. More than 35 pests are recorded on pepper of which, aphids, thrips and mites are the most important [15]. Systemic and contact insecticides belonging to the organophosphate, synthetic pyrethroid and neonicotinoid groups are being used to contain the pest complex because of their excellent biological activities and high effectiveness [16,17].

Pesticide residues are important contaminants and a trade barrier for the export and import of the agricultural commodities across the globe and indicators of pollution in the environment [18,19,20,21], leading to potential health impacts on living beings [22]. Monitoring of pesticide residues in vegetables requires high sensitive and reproducible methods. The quick, easy, cheap, effective, rugged and safe (QuEChERS) method of sample extraction has been widely adopted for the multiresidue analysis of different matrices [23,24,25]. Application of modified QuEChERS methods in peppers to determine pesticideresidues was reported [13,26,27,28], including analysis of tricyclazole in rice [29] chlorantraniliprole in pigeonpea [30]; indoxacarb in pigeonpea [31]; profenofos in pigeonpea [32]; multiple residues in pigeonpea [33]. The detection of pyridaben in unprocessed and processed peppers [15]; etoxazole [34] and a group of 16 pesticides [35] in hot peppers using LC-MS/MS are a few reports of suitable methods.

The presence of pesticide residues is a major bottleneck in the international trade for the exchange of food commodities [36]. To reduce consumer exposure, legislation has been enacted to reduce the levels of harmful pesticides and appropriate use with the established maximum residue limits (MRLs) [37,38]. Monitoring of pesticides following efficient extraction methods and determination with a sensitive technique such as liquid chromatography-tandem mass spectrometry (LC-MS/MS) and gas chromatography-tandem mass spectrometry (GC-MS/MS) could meet the regulatory requirements [38] for focusing on the proper use of pesticides with regard to application rates following Good Agricultural Practices (GAP) and safe waiting periods [36,39,40]. With this scientific background, the present study was conducted to develop a sensitive and reproducible analytical method for the simultaneous determination of 77 multiclass pesticides and their metabolites in tomato and *Capsicum*, and estimation of measurement uncertainties associated with the proposed validated method.

## 2. Results and Discussions

### 2.1. Optimization of Instruments Parameters

The analytical parameters for 77 different chemical pesticides were optimized with the given analytical conditions for the GC-MS/MS and LC-MS/MS technique. All the tested pesticides were separated with good resolution by the LC-ESI-MS/MS and GC-EI-MS/MS chromatographic determination (Figure 1). The chromatographic conditions for LC-MS/MS were optimized for better separation of pesticide mixtures in the *Capsicum* and tomato matrix. The mobile phase with methanol provided better ionization for the tested pesticides. A C_18_ column (octadecylsilyl, III; 2 mm i.d × 150 mm × 2.2 µm) was used for chromatographic separation and a gradient program of 25 min. duration produced a better separation with good peak shape for 39 pesticides in LC-MS/MS. Full scan mass spectra of 39 different pesticides were used to select the most abundant mass-to-charge (*m/z*) ions. The selection of a minimum of three different ions (for confirmation and quantification) for all the test analytes fulfilled the requirements as per the SANTE/12682/2019 [41]. For the analytes, the protonated molecular ions (M + H)^+^ were determined and chosen as the precursor ions (Table 1). The multiple reaction monitoring (MRM) transitions and associated acquisition parameters were optimized for the maximum abundance of the fragmented ions under ESI positive mode by injecting 2 µL of a 0.1 µg mL^−1^ pesticides standard mixture into the tandem mass spectrometer. Then dissociation was induced using argon gas and different collision energies were tested to find the most abundant product ions. The optimized precursor *m/z* and product ion transitions with CE were used for quantification of different pesticide residues in real samples. The developed LC-multiple reaction monitoring (MRM) mode provides high sensitivity and selectivity requirements for the analytical method used for the detection of multiclass pesticides at the lowest concentration in the *Capsicum* and tomato matrices (Table 1 and Figure 2).

The GC-MS/MS acquisition parameters such as precursor, product ions and corresponding collision energy (CE) were selected and optimized. Initially, the complete precursor and product ion scan for each pesticide was investigated and then the collision energy was optimized to obtain the best response for two selected product ions. Theoretically, the best option for the choice of precursor ion for MS/MS fragmentation is the base ion in the mass spectrum as it shows the highest intensity. The MRM transitions (parent ions and product ions) with the corresponding collision energy (CEs) and retention time (RT) given in Table 1 were used for identification and effective separation with good resolution of the peaks. The multiple reaction monitoring (MRM) method was used for monitoring different product ions of 38 different chemical pesticides in *Capsicum* and tomato samples. Suitable MRM transitions for each pesticide were selected carefully to ensure specificity and simultaneous determination of the 38 pesticide compounds in *Capsicum* and tomato matrices.

### 2.2. Method Verification in Capsicum

The method validation parameters are summarized in Table 2 and Table 3 for *Capsicum* and tomato, respectively. The calibration curve for each pesticide was linear over the concentration range from 0.01 to 1.00 µg mL^−1^ and the coefficient of determinations (R^2^) ranged from 0.995 to 0.999 for *Capsicum*. The LOD and LOQ of the method ranged from 0.19 to 10.91 and 0.63 to 35.35 µg·kg^−1^ respectively, for the *Capsicum* matrices. The limit of quantification (LOQ) was considered as the lowest spike level of analytes in a sample that could be quantified with acceptable precision and recovery of 70–120%. The LC-MS/MS and GC-MS/MS techniques were found to be highly sensitive and can quantify the pesticide residue below their Maximum Residual Limits (MRLs) established by the Food Safety Standards Authority of India and European Union for *Capsicum* (Table 2).

Accuracy of the method was assessed by spiking the *Capsicum* and tomato samples with a pesticide standard mix solution at concentrations of 50, 100 and 200 µg·kg^−1^ (Figure 3a,b).

The recovery in *Capsicum* was recorded within the range of 70.00 to 120.00% for the 77 pesticides (Table 2). In GC-MS/MS analysis pertaining to the *Capsicum* matrix recoveries in the range of 78.95 to 115.92, 79.61 to 132.13 and 81.27 to 114.93% at, 50, 100 and 200 µg·kg^−1^, were obtained, respectively, whereas, 79.39 to 111.15, 74.00 to 118.23 and 78.95 to 107.21% recovery were obtained for the for LC-MS/MS-amenable pesticides at 50, 100 and 200 µg·kg^−1^, respectively. Intraday precision test for the method was determined by spiking 100 µg·kg^−1^ to the matrix found acceptable recovery of 83.42 to 98.97% and RSD of 1.59 to 15.95% for pesticides tested by GC-MS/MS and 76.50 to 114.66% with RSD of 4.42 to 13.71% for pesticides tested by LC-MS/MS. Interday precision revealed that average recoveries of 83.03 to 105.10% and 83.96 to 113.65% for the first and second day, respectively, with RSD of 20% in GC-MS/MS, whereas 75.00 to 104.20% on the first day and 79.18 to 106.01% recovery with RSD < 20% was obtained in subsequent day analysis for the pesticides in LC-MS/MS. Recovery, repeatability and reproducibility were all found acceptable as per the SANTE/12862/2019 guidelines [41]. The percent matrix effect was calculated by comparing the angular coefficient obtained in calibration curve drawn with solvent and matrix match standard solution prepared using *Capsicum* control extracts for each pesticide. The matrix effect was in the range of 2.78 to 44.50% with the GC-MS/MS techniques and −55.00 to 32.00% for the LC-MS/MS technique. The matrix effect was found within in the limit of ±20% for all the pesticides effect for few compounds. The highest matrix-induced signal enhancement effect and the lowest matrix-induced signal suppression was recorded. 

The present method was in accordance with the methods developed for determination of pesticides residues in pepper was found acceptable interms its accuracy, repeatability and reproducibility [13,26,27,28]; analysis of tricyclazole in rice with lowest detection level (0.002 μg g^–1^) was achieved and could be useful for monitoring of rice samples subjected to export [29]; chlorantraniliprole, indoxacarb and profenofos analysis in pigeonpea following a method standardized through LC-MS/MS with lowest detection and quantification level [30,31,32] and multiple residues in pigeonpea [33]. Pyridaben analysis in unprocessed and processed peppers [13] and etoxazole [34] are promising methods to identify and quantify the corresponding residues. An LC-MS/MS method was employed to analyze 16 different pesticides in hot peppers reaching a similar conclusion with respect to proposed method [35]. Pesticide residue analysis in *Capsicum* is challenging due to its complex matrix nature and can be overcome by applying the modified QuEChERS protocol method and an acceptable recoveryobtained.

### 2.3. Method Verification in Tomato

Coefficients of determination (R^2^) ranging from 0.994 to 0.999 for tomato matrices were achieved for the tested pesticides. The calculated LOD and LOQ were in the range of 0.10 to 9.55 and 0.35 to 33.43 µg·kg^−1^, respectively, for tomato matrices. The tomato control sample fortified at 50, 100 and 200 µg·kg^−1^ and extracted following the standardized method provided recoveries in the range of 71.57 to 101.38%; 72.16 to 95.03%; 70.28 to 105.31% (Table 3 and Figure 4a). The intraday (RSD_r_) precision test recorded recoveries of 68.92 to 96.05% with a RSD less than 20%. Interday (RSD_wr_) tests recorded recoveries of 67.94 to 97.55% on day one and 71.14 to 102.84% in the second day test witha RSD of less than 20.00%. The optimized method was found suitable and reproducible with tomato matrices for the analysis of 77 different pesticides. The percent matrix effect was in the range of −18.30% to 26.77%. The highest matrix-induced signal enhancement effect was seen for triazophos (26.77%) and a signal suppression effect (−18.30%) was recorded for metalachlor. The present investigation on method development and validation is in line in terms of matrix effect, linearity, precision and accuracy, sensibility (LOD and LOQ) and repeatabilitywith a study conducted by Bozena etal. in tomatoes and cucumbers with a multiresidue analytical method. A study conducted for the analysis ofbenalaxyl, chlorothalonil and methomyl in tomato was found acceptable and recorded a lowest detectable range of 0.12 to 10 picograms in tomato [8,42].

### 2.4. Estimation of Measurement Uncertainty

The combined uncertainty occurring due to the recovery of the target analytes at 50 µg·kg^−1^ spiking levelin both matrices was in the range of 6.50 to 23.94 µg·kg^−1^ for tomato and 2.79 to 12.29 µg·kg^−1^ for *Capsicum*
Table 2 and Table 3). The uncertainty values associated with the method for *Capsicum* and tomato matrices was found acceptable. Similar results observed with estimation uncertainty during analysis of spent leaves, made tea and tea infusion was found acceptable for most of the pesticides and it indicated that the method has high sensitivity and is suitable for multi-residues analysis [41,43] and results are in accordance with estimation of uncertainty in a pigeonpea matrix analyzed with GC-MS/MS and LC-MS/MS [33].

## 3. Material and Methods

### 3.1. Chemical and Reagents

Certified reference material (CRM) was procured from Dr. Ehrenstorfer (Augsburg, Germany). LC-MS grade acetonitrile and methanol (≥99.9%) were procured from J. T. Baker (New Jersey, USA), Ammonium formate and formic acid (≥90.00%) were bought from Empart (Hyderabad, India). Ethyl acetate (≥99.9%) was procured from Merck Mumbai, India. Ultrapure water of 18.2 MΩ was obtained using a Milli-Q water purification system (Merck Millipore, Mumbai, India). Anhydrous magnesium sulphate (≥99.90%), anhydrous sodium sulphate, graphitized carbon black (GCB), anhydrous sodium chloride (≥99.90%) was obtained from Himedia (Bangalore, India). The primary secondary amine (PSA, 40 μm particle size) was obtained from Agilent Technologies, (California, CA, USA).

### 3.2. Preparation of Standard Solution

A standard stock solution was prepared by accurately weighing 10 ± 0.10 mg of CRMs into a 10 mL calibrated volumetric flask and made up to 10 mL with methanol (JT baker, New Jersey, NJ, USA) and ethyl acetate (Merck Mumbai, India) for LC and GC amenable pesticides, respectively. An intermediate and working solution of known concentration i.e., 400 and 10 µgmL^−1^, respectively was prepared following serial dilution. Further, a linear standard concentration ranging from 0.01 to 1.00 µg mL^−1^ was prepared to construct the calibration curve. By using the control *Capsicum* and tomato extract, matrix match standards were prepared. All the solutions were stored at −15 °C.

### 3.3. Extraction and Clean-Up

Ground *Capsicum* and tomato matrix (10 g) was weighed and transferred into a 50 mL centrifuge tube. To this, 20 mL of acetonitrile and 5 mL of water was added and allowed to stand for 30 min. The sample mixture was then homogenized at 10,000–13,000 rpm for 3 min. Then, 3 g of NaCl was added and vortexed immediately for 2 min. The homogenized sample extract was centrifuged at 12,000 rpm for 5 min. at 10 °C. After centrifugation, 15 mL of the upper organic layer was collected into a test tube and 9 g of sodium sulphate was added. Further, 11 mL of extract was transferred into a 15 mL centrifuge tube containing 0.4 g of primary secondary amine (PSA) and 1.15 g of magnesium sulphate and then vortexed the mixture for one minutes. Centrifuged the supernatant at 12,000 rpm for 5 min. Then, 1 mL of supernatant was filtered using 0.22 µm PTFE nylon filter into LC vials. For GC analysis, 3 mL of supernatant was collected in a test tube and concentrated using nitrogen flash evaporator and reconstituted with 1.5 mL ethyl acetate and filtered into GC vials using a 0.22 µm PTFE nylon filter. Samples were further subjected to LC-MS/MS and GC-MS/MS analysis.

### 3.4. LC-MS/MS

A LCMS 8040 series LC system (Shimadzu^®^, Kyoto, Japan) consisting of a solvent degassing unit, a binary pump, an autosampler, and a thermostated column compartment was used in the LC-MS/MS system. Separation of the analytes was achieved on a Shimpack XR ODS C_18_ column (150 × 2 mm i.d.; 2.2 µm particle diameter) with a column oven temperature of 40 °C. Mobile phase A consisted of 5 mM ammonium formate, 2 mL methanol, and 0.01% formic acid and made-up to the volume of 100 mL with HPLC water and mobile phase B consisted the 5 mM ammonium formate, 0.01% formic acid and made up of the volume of 100 mL with methanol. Q1 scanning at the mobile phase flow rate of 0.4 mL min^−1^ was performed with the union joint and binary gradient. The standard *m*/*z* (+) were selected based on full scan mass spectra of pesticide compounds with electrospray ionization positive mode (ESI^+^). Further, the protonated molecular ions (M + H) were determined and chosen as the precursor ions. After that, the MRM method was created to select the product ions. The optimization of the method was done in an MRM positive mode with 1.00 min acquisition time (seven scans) and collision energy upper and lower limits of −50 to 0 and 0 to +50 was adopted with an injection volume of 2 µL of 0.1 µg mL^−1^ standard. The MS source parameters used were as follows: interface voltage of 4.5 kV, desolvation temperature of 250 °C, heat block temperature of 400 °C, desolvation gas (N_2_) of 2.9 L·min^−1^ and drying gas at 2.9 L·min^−1^.

### 3.5. GC-MS/MS

The extracts were analyzed using gas a chromatograph (Shimadzu, GC 2100, Shimadzu^®^, Kyoto, Japan) equipped with a MS system (TQ 8030) coupled with an electronic flow controller (EFC), an AOC 20i injector and AOC 20S auto sampler was used for instrument operation. Data acquisition and processing was controlled through LabSolution^®^ software version 5. The injection volume was 2 µL in split less mode, surge pressure: 250 kpa (1 min) and injector temperature was maintained at 250 °C. A capillary fused silica HP 5 MS column (30 m × 0.25 mm with 0.25 µm film thickness) was used to separate the target analytes. The column oven temperature program was 60 °C (held for 1 min), ramped at 40 °C per minute to 170 °C (held for zero minutes) and finally ramped at 10 °C per minute to 310 °C (held for 3 min) with a run time of 36 min. Carrier gas was helium (99.999% purity) with constant flow rate of 1.4 mL min^−1^ was maintained. MS parameters such as transfer line temperature and ion source temperature was of 280 and 250 °C, respectively was maintained. Electron impact (EI) ionization with positive mode using electron energy of –70 eV. The MRM scan mode was selected. The solvent delay was fixed at 3 min. Argon (Ar) and helium (He) gases with flow rates of 1.50 and 2.25 mL min^−1^ were used as collision and quench gases, respectively. The mass range was between 50–550 *m*/*z*.

### 3.6. Method Verification

For verification of the method, about 1kg each *Capsicum* and tomato sample (pesticide-free) was collected and processed (Figure 4a,b). The matrix sample was extracted as per the procedure explained in the extraction and cleanup section. Identification and quantification of multi-class pesticides residues in capsicum and tomato was optimized and validated according to the SANTE/12682/2019guidelines [41]. Linearity, matrix effect, limit of detection (LOD) and quantification (LOQ), specificity, trueness (bias), precision in terms of repeatability (RSDr-intraday), and precision in terms of reproducibility (RSD_wR_-interday), ion ration and retention time were evaluated. The standard concentrations ranged between 0.01 to 1.00 µg mL^−1^ prepared in solvent and matrix and coefficient of determination was calculated. Matrix effect was calculated by comparing the angular coefficients recorded in the calibration curves drawn using the solvent and matrix linearity test. The formulae used for calculation of matrix effect as follows:Matrix effect (%) = (b_m_ ‒ b_s_)/b_s_ × 100(1)
where, b_m_ and b_s_ are the angular coefficients of the curve in the matrix and in the solvent, respectively [32,44]. The LOD was calculated by injecting the lowest concentration which is expected to produce a response. LOQs were estimated as the concentration of pesticide that produces recovery of lowest spike level within the limit of 70–120% with RSD of <20%.

The recovery study was conducted by spiking the *Capsicum* and tomato samples with the pesticide standard mixture solution at three fortification levels viz., 50, 100 and 200 µg·kg^−1^ with six replications for each level. To allow the absorption of pesticides compounds into the matrix, the sample matrix were kept at room temperature (25 °C) for 2 h after fortification. The method accuracy was determined concerning the repeatability with relative standard deviation (RSDr) similarly extractions of blank capsicum and tomato matrix spiked with pesticides at the same fortification levels (50, 100, 200 µg·kg^−1^). RSD concerning reproducibility (RSD_wR_) was estimated by conducting the fortification study at two different days (r = 6) with similar fortification level. A minimum of three transitions (*m*/*z*) (one precursor and two product ions) were selected and checked for their ion ratio (less than 30%) in the method.

### 3.7. Estimation of Measurement Uncertainty

The expanded uncertainty (U_exp_) at 95% confidence limit was calculated at 50 µg·kg^−1^ of spiking level. The statistical methods mentioned in the EURACHEM/CITAC Guide was followed to estimate the measurement of uncertainty in food samples [43,45]. The procedure explained for estimation of measurement uncertainty in pigeonpea matrix was followed for calculation of measurement uncertainty for intra laboratory validation (for recovery) of multiresidue residue analytical method for *Capsicum* and tomato [33].

## 4. Conclusions

Finally, it is concluded that the analytical methods for simultaneous determination of 77 multi-class pesticides residues in *Capsicum* and tomatoes using LC-MS/MS and GC-MS/MS is highly sensitive, and can detect and quantify the residues below their prescribed EU and FSSAI MRLs. The proposed optimized methods are suitable for screening of 77 different chemical pesticides in vegetable matrices in a short time (less than 40 min). The methods have linear regression coefficients in the range of 0.994 to 0.999. Combined uncertainty (Uc) and expanded uncertainty (U_exp_) were found accurate and within the limits. This method has high accuracy (between 79.39 to 115.92%), precision, reproducibility and ruggedness, making it suitable for adoption in large scale monitoring of vegetables collected from farmers’ fields and markets.

## Figures and Tables

**Figure 1 molecules-26-01837-f001:**
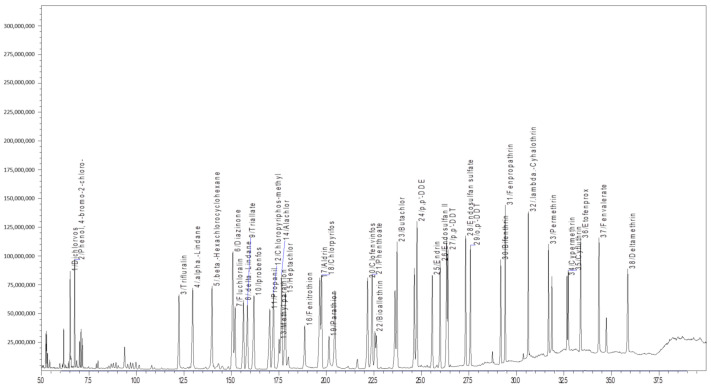
GC-MS/MS chromatogram for multiclass pesticides at 0.1 mg kg^−1^.

**Figure 2 molecules-26-01837-f002:**
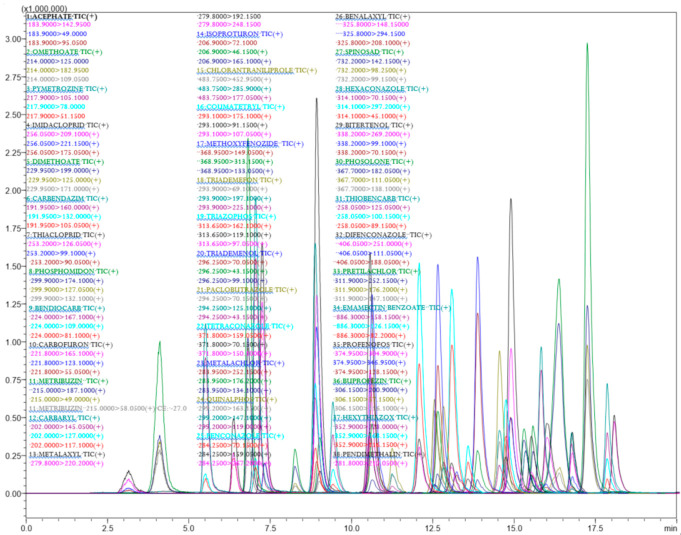
LC-MS/MS chromatograms of 39 pesticides in 25 min run showing different MRM transitions.

**Figure 3 molecules-26-01837-f003:**
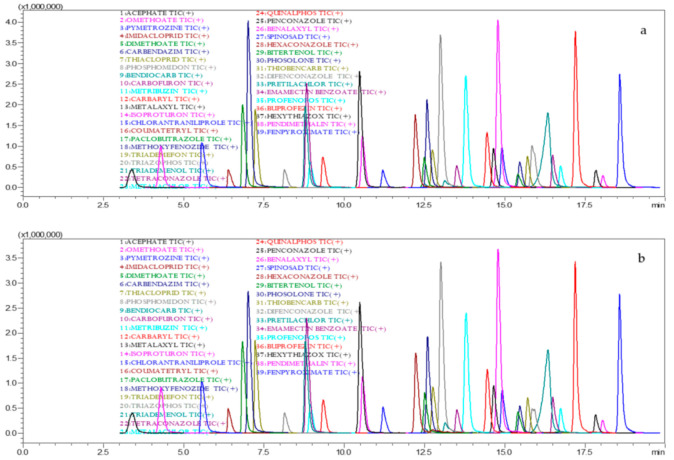
LC-MS/MS chromatogram for recovery in capsicum (**a**) and tomato (**b**) at 0.1 mg kg^−1.^

**Figure 4 molecules-26-01837-f004:**
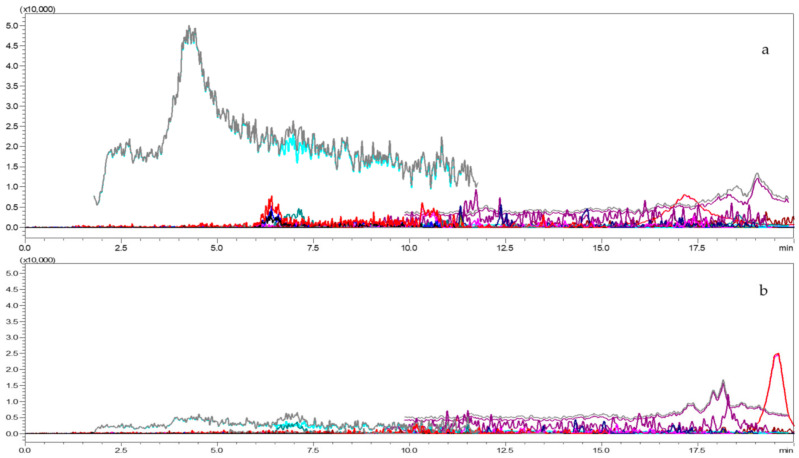
LC-MS/MS chromatograms for blank tomato (**a**) and *Capsicum* matrix (**b**).

**Table 1 molecules-26-01837-t001:** Retention time, mass spectrometric parameters for multiclass pesticides analyzed in tomato and *Capsicum*.

Pesticide	RT (Min.)	Quantification		Confirmation			
		MRM Transition (*m*/*z*)	Collision Energy (eV)	MRM Transition (*m*/*z*)	Collision Energy (eV)	MRM Transition(*m*/*z*)	Collision Energy (eV)
**GC-MS/MS**							
Dichlorvas	6.41	185.0 > 93.0	14	185.0 > 109.0	14	185.0 > 63.0	22
4-Bromo-2-chloromophenol,	6.56	208.0 > 63.10	27	208.0 > 99.10	21	208.0 > 144.00	15
Trifluralin	12.34	110.1 > 64.0 18	18	152.1 > 110.1	8	110.1 > 92.0	12
α-BHC	12.81	306.1 > 264.1	8	306.1 > 206.1	14	264.1 > 206.1	8
β-BHC	14.04	218.9 > 182.9	8	180.9 > 144.9	16	180.9 > 74.0	30
Diazinon	15.17	304.1 > 179.1	10	304.1 > 162.1	8	304.1 > 137.1	26
Fluchloralin	15.38	306.0 > 264.10	6	306.0 > 160.20	27	306.0 > 206.20	18
Delta-BHC	15.36	218.9 > 182.9	10	218.9 > 144.9	20	180.9 > 144.9	16
Tri-allate	15.67	268.1 > 226.0	14	268.1 > 184.0	20	270.1 > 228.0	14
Iprobenfos	16.06	204.0 > 91.0	8	204.0 > 171.0	6	123.0 > 45.0	16
Propanil	16.87	160.9 > 99.0	24	217.0 > 161.0	10	217.0 > 57.0	20
Chlorpyrifos-methyl	17.31	285.9 > 93.0	22	287.9 > 272.9	20	287.9 > 93.0	22
Parathion-methyl	17.30	125.0 > 47.0	12	263.0 > 109.0	14	125.0 > 62.0	6
Alachlor	17.73	188.1 > 160.1	10	188.1 > 132.1	18	160.1 > 132.1	10
Heptachlor	17.50	271.8 > 236.9	20	271.8 > 117.0	32	271.8 > 201.9	38
Fenitrothion	18.77	277.0 > 260.0	6	277.0 > 109.1	14	260.0 > 125.1	22
Aldrin	19.20	292.9 > 219.9	26	292.9 > 257.9	16	292.9 > 186.0	40
Chlorpyrifos	19.91	313.9 > 257.9	14	313.9 > 285.9	8	313.9 > 193.9	28
Ethyl parathion	19.95	291.1 > 109.0	14	291.1 > 137.0	6	291.1 > 81.0	24
Chlorfenvinphos	22.21	267.0 > 159.0	18	323.0 > 267.0	16	323.0 > 295.0	6
Phenthoate	22.34	119.1 > 82.1	28	119.1 > 84.1	28	149.1 > 105.1	4
Allethrin	22.39	273.9 > 125.0	20	246.0 > 121.0	6	246.0 > 63.0	28
Butachlor	23.76	188.1 > 160.1	12	188.1 > 132.1	18	176.1 > 134.1	12
*p,p’*-DDE	24.53	246.0 > 176.0	30	246.0 > 211.0	22	317.9 > 248.0	24
Endrin	25.22	262.9 > 193.0	28	262.9 > 228.0	22	244.9 > 210.0	8
β-Endosulfan	25.62	338.9 > 160.0	18	338.9 > 266.9	8	338.9 > 195.9	20
*p,p’*-DDT	26.12	235.0 > 165.0	24	235.0 > 199.0	16	237.0 > 199.0	16
Endosulfan sulfate	27.13	386.8 > 288.8	10	386.8 > 252.9	16	386.8 > 240.9	22
*o,p’*-DDT	27.34	235.0 > 165.0	24	235.0 > 199.0	16	237.0 > 199.0	16
Bifenthrin	29.08	181.1 > 166.1	12	181.1 > 153.1	8	181.1 > 179.1	12
Fenpropathrin	29.23	265.1 > 210.1	12	265.1 > 89.0	28	181.1 > 127.1	28
λ-Cyhalothrin	30.55	181.1 > 152.1	24	163.1 > 127.0	14	-	-
Permethrin	31.49	163.1 > 127.1	8	183.1 > 165.1	14	183.1 > 153.1	14
Cypermethrin	30.55	163.1 > 91.0	14	181.1 > 152.1	22	181.1 > 127.1	22
Cyfluthrin	32.56	226.1 > 206.1	14	226.1 > 199.1	6	163.1 > 91.0	14
Etofenprox	33.16	163.1 > 135.1	10	163.1 > 107.1	18	163.1 > 95.0	18
Fenvalerate	34.20	419.1 > 225.1	6	419.1 > 167.1	12	419.1 > 125.1	26
Deltamethrin	35.64	252.9 > 93.0	20	252.9 > 171.9	8	252.9 > 77.0	26
**LC-MS/MS**							
Acephate	1.06	183.9 > 142.95	11	183.9 > 49	23	183.9 > 95.05	26
Omethoate	1.09	214 > 125	24	214 > 182.95	12	214 > 109.05	30
Pymetrozine	1.32	217.9 > 105.1	22	217.9 > 78	44	217.9 > 51.15	54
Imidacloprid	2.03	256.05 > 209.1	17	256.05 > 221.15	9	256.05 > 175.05	19
Dimethoate	2.67	229.95 > 199	10	229.95 > 125	22	229.95 > 171	17
Carbendazim	3.05	191.95 > 160	18	191.95 > 132	32	191.95 > 105.05	40
Thiacloprid	3.22	253.2 > 126.05	23	253.2 > 99.1	48	253.2 > 90.05	40
Phosphomidon	4.65	299.9 > 174.1	14	299.9 > 127.05	28	299.9 > 132.1	24
Bendiocarb	5.42	224 > 167.1	11	224 > 109	19	224 > 81.1	35
Metribuzin	5.43	215 > 187.1	20	215 > 49	29	215 > 58.05	27
Carbofuron	5.47	221.8 > 165.1	12	221.8 > 123.1	23	221.8 > 55.05	29
Carbaryl	6.12	202 > 145.05	12	202 > 127	28	202 > 117.1	24
Isoproturon	7.48	206.9 > 72.1	23	206.9 > 46.15	18	206.9 > 165.1	15
Metalaxyl	7.52	279.8 > 220.2	15	279.8 > 192.15	18	279.8 > 248.15	11
Chlorantraniliprole	8.31	483.75 > 452.95	17	483.75 > 285.9	16	483.75 > 177.05	50
Coumatetryl	9.11	293.1 > 175.1	24	293.1 > 91.15	36	293.1 > 107.05	35
Paclobutrazole	9.47	294.25 > 70.15	23	294.25 > 125.1	37	294.25 > 43.15	50
Methoxyfenozide	9.63	368.95 > 149.05	18	368.95 > 313.15	9	368.95 > 133.05	24
Triademefon	9.72	293.9 > 69.1	23	293.9 > 197.1	17	293.9 > 225.1	14
Triazophos	9.98	313.65 > 162.1	19	313.65 > 119.1	35	313.65 > 97.05	37
Triademenol	10.04	296.25 > 70.05	12	296.25 > 43.15	47	296.25 > 99.1	18
Tetraconazole	10.4	371.8 > 159.05	34	371.8 > 70.15	25	371.8 > 150.1	35
Metalachlor	10.64	283.95 > 252.15	15	283.95 > 176.2	27	283.95 > 134.1	34
Quinalphos	11.2	299.2 > 163.15	22	299.2 > 147.1	22	299.2 > 119	44
Penconazole	11.37	284.25 > 70.15	18	284.25 > 159.05	31	284.25 > 267.2	8
Benalaxyl	11.55	325.8 > 148.15	23	325.8 > 294.15	12	325.8 > 208.1	16
Hexaconazole	11.86	314.1 > 70.15	22	314.1 > 297.2	10	314.1 > 45.1	39
Bitertenol	12.03	338.2 > 269.2	10	338.2 > 99.1	17	338.2 > 70.15	11
Phosalone	12.05	367.7 > 182.05	16	367.7 > 111.05	43	367.7 > 138.1	32
Spinosad	12.17	732.2 > 142.15	35	732.2 > 98.25	54	732.2 > 99.15	52
Thiobencarb	12.25	258.05 > 125.05	20	258.05 > 100.15	13	258.05 > 89.15	49
Difenconazole	12.46	406.05 > 251	28	406.05 > 111.05	55	406.05 > 188.05	47
Pretilachlor	12.81	311.9 > 252.15	16	311.9 > 176.2	29	311.9 > 147.1	40
Profenofos	13.13	374.95 > 304.9	21	374.95 > 346.95	14	374.95 > 128.15	47
Emamectin benzoate	13.31	886.3 > 158.15	42	886.3 > 126.15	48	886.3 > 82.2	50
Buprofezin	13.51	306.15 > 200.9	11	306.15 > 57.15	30	306.15 > 116.1	16
Hexythiazox	14.00	352.9 > 228	15	352.9 > 168.15	28	352.9 > 116.15	43
Pendimethalin	14.18	281.8 > 212.05	11	281.8 > 199.95	8	281.8 > 193.95	20
Fenpyroximate	14.62	422 > 366.2	16	422 > 138.2	35	422 > 215.15	27

**Table 2 molecules-26-01837-t002:** Coefficient of determination (*R^2^*), LOD, LOQ, recovery and repeatability of multiclass pesticide analyzed in capsicum using GC-MS/MS and LC-MS/MS.

Compounds	R^2^	LOD(µg/kg)	LOQ(µg/kg)	Recovery (%)			Repeatability (100 µg·kg^−1^)		Reproducibility (100 µg·kg^−1^)			Matrix Effect (%)	MU at 50 µg·kg^−1^
				50(µg/kg)	100 (µg/kg)	200 (µg/kg)	Recovery (%)	RSD (%)	Recovery		RSD(%)		
									Day 1	Day 2			
**GC-MS/MS**													
Dichlorvos	0.999	3.40	11.34	94.07	90.04	94.74	89.95	5.92	90.71	89.90	2.58	7.69	±7.34
4-Bromo-2-chlorophenol	0.997	0.27	0.91	83.27	86.22	89.51	90.37	5.20	88.66	77.94	4.02	5.09	±6.82
Trifluralin	0.999	7.28	24.28	104.32	96.24	90.94	91.09	4.67	102.03	112.05	8.03	4.77	±5.79
α-BHC	0.999	1.95	6.51	103.27	96.40	98.91	92.65	3.92	96.48	95.96	4.81	5.37	± 5.12
β-BHC	0.999	2.90	9.65	103.11	94.35	92.70	90.81	5.05	92.56	95.04	7.62	4.88	±5.76
Diazinon	0.999	1.20	3.99	113.35	91.28	94.53	95.45	8.02	99.84	111.08	8.96	8.56	±5.77
Fluchloralin	0.997	3.19	10.62	103.94	102.43	92.27	90.88	5.83	105.10	99.67	11.63	2.87	±2.79
delta-BHC	0.999	2.30	7.66	102.62	98.51	91.16	91.23	5.13	93.49	100.35	6.58	4.53	±4.35
Tri-allate	0.997	1.50	5.02	108.10	95.89	98.37	96.92	5.55	90.60	105.83	12.45	25.99	±5.76
Iprobenfos	0.997	7.06	23.52	111.19	102.57	95.78	98.97	5.97	98.49	108.92	8.57	6.45	±6.59
Propanil	0.997	8.34	27.80	86.23	84.71	82.33	93.10	3.78	83.03	83.96	6.53	11.12	±5.24
Chlorpyrifos-methyl	0.997	1.06	3.53	101.65	90.69	91.86	90.46	5.18	95.32	99.38	12.35	9.94	±3.71
Parathion-methyl	0.997	4.13	13.78	99.54	95.82	91.81	97.17	5.73	98.16	97.27	8.58	8.40	±5,74
Alachlor	0.998	4.41	14.72	111.35	96.72	101.49	96.15	5.48	99.55	99.08	7.91	5.79	±6.59
Heptachlor	0.997	2.76	9.20	115.92	105.44	99.12	97.90	4.78	91.29	113.65	16.81	11.87	±2.83
Fenitrothion	0.998	1.35	4.50	105.80	98.85	112.83	91.41	4.83	101.06	103.53	3.24	2.92	±7.49
Aldrin	0.997	1.74	5.80	103.01	132.13	110.52	99.77	7.58	96.32	100.74	4.75	7.69	±5.63
Chlorpyrifos	0.999	10.91	36.35	108.09	90.49	90.48	92.55	3.88	89.90	105.82	12.99	4.14	±5.45
Parathion-ethyl	0.999	2.90	9.67	102.98	103.94	98.60	94.72	5.00	95.40	100.71	5.40	10.20	±7.26
Chlorfenvinphos	0.999	8.92	31.22	107.59	97.89	91.76	90.08	5.22	93.50	105.32	9.91	22.08	±4.62
Phenthoate	0.997	3.17	10.58	108.54	94.23	93.52	98.79	5.94	96.61	106.27	8.22	12.50	±5.38
Allethrin	0.996	0.19	0.63	103.60	106.66	98.60	98.02	1.59	99.06	101.33	3.17	7.51	±6.53
Butachlor	0.998	10.17	33.91	105.83	106.57	102.63	92.62	3.89	96.81	103.56	6.29	6.40	±6.29
*p,p’*-DDE	0.997	3.93	13.09	101.25	96.73	95.32	87.28	6.80	96.78	98.98	3.19	10.52	±5.01
Endrin	0.997	2.68	8.92	106.69	105.07	110.00	85.14	8.09	94.64	104.42	8.46	3.27	±3.05
β-Endosulfan	0.999	9.93	33.09	114.75	99.41	106.00	93.58	15.95	98.00	88.73	9.71	15.93	±6.63
*p,p*’-DDT	0.999	2.44	8.14	105.44	102.03	96.87	98.20	6.12	94.75	103.17	7.56	7.07	±2.23
Endosulfan sulfate	0.999	0.56	1.88	115.79	97.90	98.56	83.42	12.80	95.56	113.52	11.23	13.33	±11.71
*o,p’*-DDT	0.997	2.23	7.44	105.80	97.84	114.93	98.41	5.63	98.74	103.53	13.66	2.78	±4.24
Bifenthrin	0.998	1.5	4.90	101.68	93.28	98.14	90.94	4.95	96.73	99.41	3.53	7.13	±9.14
Fenpropathrin	0.997	6.09	20.30	104.63	95.93	93.81	92.68	6.44	93.68	102.36	7.81	10.50	±4.61
λ-Cyhalothrin	0.999	8.05	26.84	96.76	89.74	98.92	89.97	5.62	99.66	94.49	2.09	23.96	±12.09
Permethrin	0.999	2.57	8.55	92.01	88.34	93.51	89.21	6.15	91.52	89.74	8.37	9.63	±2.87
Cypermethrin	0.995	5.07	17.74	95.89	87.41	90.69	90.26	5.50	98.57	93.62	1.95	34.33	±8.14
Cyfluthrin	0.999	1.45	4.84	78.95	79.61	81.27	90.28	5.79	90.83	88.68	12.82	32.63	±3.47
Etofenprox	0.998	1.55	5.17	86.93	85.18	86.47	90.40	5.37	84.30	84.66	3.01	5.51	±2.97
Fenvalerate	0.996	1.61	5.35	89.14	84.64	90.05	86.77	7.83	88.69	94.87	16.51	37.50	±7.55
Deltamethrin	0.998	7.95	26.50	96.19	84.04	88.39	93.81	3.98	94.67	93.92	1.13	44.50	±8.52
**LC-MS/MS**													
Thiacloprid	0.999	2.65	8.84	83.43	83.32	85.37	88.66	4.90	81.77	92.50	7.41	4.3	±8.53
Buprofezin	0.999	2.50	8.32	85.22	87.56	91.28	83.28	5.08	88.84	94.62	5.63	−1.6	±10.91
Metalachlor	0.999	2.59	8.64	83.25	89.32	80.55	80.16	4.42	91.05	84.50	7.38	23.0	±8.73
Imidacloprid	0.999	5.66	18.87	88.82	83.00	85.18	84.62	9.40	97.87	95.12	10.50	1.6	±8.68
Dimethoate	0.999	1.16	3.88	79.39	85.13	83.05	90.14	6.67	86.85	93.16	5.92	−2.5	±12.20
Coumatetryl	0.999	1.24	4.14	85.86	89.00	82.38	84.68	8.39	93.61	79.18	8.09	6.1	±3.56
Triademenol	0.999	9.32	32.62	91.64	87.00	106.97	87.62	7.26	104.20	83.12	6.37	−3.7	±10.66
Triademefon	0.999	2.03	6.76	85.00	102.00	90.25	99.51	4.91	97.43	90.02	5.51	−22.1	±7.84
Thiobencarb	0.998	8.52	28.41	86.00	88.12	82.12	83.10	13.71	87.89	90.54	10.94	19.0	±12.29
Spinosad	0.999	0.83	2.75	104.12	110.23	98.20	108.23	5.29	89.85	98.23	6.50	−5.6	±10.03
Phosalone	0.999	5.20	17.35	85.61	88.19	83.80	83.21	7.47	87.29	95.66	8.70	−14.5	±10.96
Methoxyfenozide	0.999	0.96	3.21	111.15	118.23	91.52	108.13	5.14	93.38	100.08	6.64	−4.8	±10.32
Hexythiazox	0.999	5.71	19.04	89.21	94.00	87.92	96.56	7.35	85.19	90.57	9.04	−22.0	±9.22
Fenpyroximate	0.999	2.07	6.90	89.56	107.00	98.83	101.00	4.93	90.03	84.68	4.78	−4.7	±11.16
Carbendazim	0.998	1.84	6.13	98.23	106.00	107.21	102.62	10.81	93.38	104.13	9.83	5.1	±10.24
Carbaryl	0.999	3.45	11.51	94.07	98.00	89.08	76.50	6.73	83.32	88.54	8.18	−9.1	±7.47
Triazophos	0.999	1.98	6.60	95.61	85.00	88.79	97.50	6.16	84.64	95.04	8.38	−2.5	±12.01
Carbofuron	0.999	1.44	4.79	85.61	83.03	88.11	81.13	4.25	96.61	90.80	5.40	−0.9	±8.91
Bitertenol	0.999	9.14	30.47	97.93	102.12	95.55	92.17	10.21	96.48	86.65	8.18	−10.6	±11.44
Bendiocarb	0.999	1.84	6.12	83.77	82.23	80.89	91.62	6.76	94.51	86.52	6.87	−41.2	±7.09
Benalaxyl	0.999	0.42	1.40	89.41	84.01	82.71	81.45	5.14	87.44	87.95	7.23	−55.0	±7.96
Acephate	0.998	3.17	10.58	87.82	85.02	78.95	87.07	10.85	81.66	100.07	9.44	−7.4	±8.65
Pymetrozine	0.999	1.55	5.16	79.47	78.12	85.14	82.68	9.63	94.55	96.28	9.99	−24.2	±11.66
Omethoate	0.999	2.68	8.92	89.61	98.12	83.92	92.61	13.24	81.70	101.11	11.26	−0.7	±12.22
Metribuzin	0.998	8.24	27.48	93.13	111.05	94.14	100.48	8.14	84.90	91.52	8.57	−10.4	±10.75
Metalaxyl	0.999	0.74	2.47	93.74	93.01	94.89	106.01	5.02	93.60	103.51	8.01	21.2	±10.94
Emamectin benzoate	0.999	0.68	2.26	92.12	98.02	96.01	95.01	4.85	89.50	93.52	3.27	−27.7	±12.18
Tetraconazole	0.999	1.10	3.68	97.39	98.12	95.40	98.56	4.70	96.82	92.01	4.41	−3.9	±10.75
Quinalphos	0.999	2.81	9.36	86.19	74.00	91.22	89.50	5.73	96.07	102.02	4.50	−34.7	±9.24
Profenofos	0.999	2.57	8.56	95.34	109.02	101.18	114.01	5.64	96.15	98.56	5.67	−10.2	±11.93
Phosphomidon	0.998	3.77	12.57	101.13	110.23	97.88	108.73	6.34	75.00	98.30	5.58	−8.2	±10.37
Pendimethalin	0.999	3.51	11.69	99.23	102.89	100.04	110.54	9.09	90.99	101.41	9.94	−1.9	±10.38
Difenconazole	0.999	1.50	5.01	94.25	101.08	86.85	114.66	5.25	92.55	106.01	6.12	−17.9	±7.61
Pretilachlor	0.999	0.62	2.08	102.17	96.00	99.67	89.00	5.82	89.87	102.00	7.92	6.3	±8.86
Paclobutrazole	0.999	3.33	11.10	97.64	89.34	93.50	96.53	6.58	102.29	104.42	7.00	32.0	±10.59
Chlorantraniliprole	0.999	1.26	4.21	98.33	88.34	96.48	89.57	8.69	87.48	100.33	9.40	2.3	±10.64
Isoproturon	0.999	5.36	17.88	88.04	89.54	88.71	88.58	6.20	85.81	98.90	7.39	−8.0	±10.51
Hexaconazole	0.998	8.01	26.71	103.34	84.57	97.84	98.75	6.07	92.56	99.08	9.40	20.2	±10.63
Penconazole	0.999	1.82	6.06	96.20	99.78	86.60	93.20	5.97	89.68	96.00	8.37	13.70	±11.57

**Table 3 molecules-26-01837-t003:** Coefficient of determination (*R^2^*), LOD, LOQ, recovery and repeatability of multiclass pesticide analyzed in tomato fruits using GC-MS/MS and LC-MS/MS.

Compounds	R^2^	LOD (µg/kg)	LOQ (µg/kg)	Recovery (%)			Repeatability (100 µg/kg)		Reproducibility (100 µg/kg)			Matrix Effect (%)	MU at 50 µg·kg^−1^ Spiking Level
				50 (µg/kg)	100 (µg/kg)	200 (µg/kg)	Recovery(%)	RSD(%)	Recovery		RSD (%)		
									Day 1	Day 2			
**GC-MS/MS**													
Dichlorvos	0.999	7.88	27.58	84.95	86.05	97.42	80.37	4.17	79.39	89.47	0.80	−1.52	±13.51
4-Bromo-2-chloro-phenol	0.999	5.36	18.76	79.57	78.93	93.25	73.05	4.76	82.07	84.02	1.58	10.95	±9.61
Trifluralin	0.999	4.11	14.39	86.16	89.96	100.85	82.05	5.27	81.07	89.57	2.17	−4.39	±10.13
α-BHC	0.999	2.63	9.21	84.57	85.81	99.19	71.30	5.90	70.32	86.58	0.72	−1.64	±8.33
β-BHC	0.996	5.15	18.03	81.98	86.43	83.03	73.42	5.87	72.44	85.93	2.38	−2.17	±8.30
Diazinon	0.999	4.43	15.51	80.29	82.14	99.06	72.82	6.87	71.84	89.47	1.33	7.95	±9.62
Fluchloralin	0.999	2.35	8.23	80.56	84.87	84.86	78.16	7.42	77.18	87.35	6.44	0.04	±8.34
delta-BHC	0.999	4.53	15.86	82.04	87.94	90.35	72.01	5.27	71.03	90.40	1.35	−1.50	±23.25
Tri-allate	0.999	3.39	11.87	82.44	83.14	89.06	75.27	7.09	74.29	88.25	0.96	−0.75	±8.08
Iprobenfos	0.999	7.32	25.62	85.02	85.66	94.07	68.92	5.08	67.94	93.78	0.88	−1.02	±10.63
Propanil	0.998	7.85	27.48	79.05	86.94	80.67	73.75	7.91	82.77	86.79	0.10	11.81	± 6.50
Chlorpyrifos-methyl	0.998	6.36	22.26	80.26	83.50	92.93	85.09	4.41	84.11	85.41	0.34	2.60	±13.18
Parathion-methyl	0.999	5.39	18.87	76.15	72.18	78.55	81.82	5.49	80.84	79.32	2.92	−4.56	±13.95
Alachlor	0.999	3.56	12.46	83.92	82.91	91.82	71.52	5.78	70.54	80.18	1.34	−2.90	±9.25
Heptachlor	0.999	5.99	20.97	79.31	87.49	97.35	76.50	4.40	75.52	89.40	2.63	11.97	±9.10
Fenitrothion	0.999	3.74	13.09	83.29	84.90	93.92	86.65	7.42	85.67	83.34	3.60	−4.86	±11.91
Aldrin	0.994	5.53	19.36	86.46	88.16	99.05	78.87	7.15	77.89	96.56	2.10	−1.90	±9.99
Chlorpyrifos	0.999	9.55	33.43	85.92	83.70	97.28	72.98	4.86	72.00	94.98	1.91	−4.00	±11.15
Parathion-ethyl	0.999	1.33	4.66	82.81	82.94	98.39	69.33	6.65	78.35	87.00	2.42	1.57	±10.64
Chlorfenvinphos	0.999	5.14	17.99	81.63	84.05	89.54	69.57	6.00	88.59	86.99	2.67	−0.48	±8.16
Phenthoate	0.999	3.90	13.65	84.35	83.02	97.82	70.29	5.42	69.31	91.24	0.95	0.64	±7.46
Aallethrin	0.997	0.10	0.35	78.49	79.93	76.24	92.94	5.60	91.96	93.24	0.22	0.15	±8.34
Butachlor	0.998	1.47	5.15	88.90	86.00	92.23	75.69	6.36	74.71	83.40	2.17	−3.33	±8.25
*p,p’*-DDE	0.999	4.54	15.89	87.98	85.70	96.75	79.87	4.68	78.89	78.74	0.12	−0.43	±8.94
Endrin	0.998	2.24	7.84	86.85	88.33	88.77	69.62	7.20	68.64	82.55	0.06	−2.27	±8.40
β-Endosulfan	0.999	2.08	7.28	94.10	89.24	87.85	77.05	15.34	76.07	97.73	9.10	5.60	±7.37
*p,p’*-DDT	0.999	5.97	20.90	85.89	86.70	92.29	80.08	4.09	79.10	86.24	0.14	−2.08	±8.24
Endosulfansulfate	0.998	1.56	5.46	83.59	81.11	88.48	76.47	15.04	85.49	80.68	4.34	2.76	±11.95
*o,p*’-DDT	0.998	8.79	30.77	80.82	82.05	74.10	73.22	5.03	72.24	78.91	5.29	6.66	±9.34
Bifenthrin	0.999	6.30	22.05	82.91	85.37	96.21	83.90	3.51	82.92	85.45	0.38	0.89	±9.21
Fenpropathrin	0.999	9.25	32.38	84.19	84.19	96.53	81.09	4.36	80.11	89.68	1.24	3.28	±8.08
λ-Cyhalothrin	0.999	5.06	17.71	89.81	92.18	99.89	83.15	3.99	82.17	85.82	0.28	3.47	±13.50
Permethrin	0.999	5.33	18.66	88.23	94.06	103.32	79.56	4.68	78.58	84.76	0.18	1.01	±10.41
Cypermethrin	0.999	6.25	21.88	81.89	82.79	98.62	83.25	4.24	82.27	81.75	1.29	0.85	±9.30
Cyfluthrin	0.997	5.20	18.20	82.89	81.97	91.32	80.49	4.11	79.51	96.54	0.84	2.54	±9.00
Etofenprox	0.999	3.84	13.44	82.47	83.04	93.46	71.78	2.99	70.80	94.56	0.22	−0.55	±12.57
Fenvalerate	0.998	2.47	8.65	85.19	82.32	97.18	79.22	8.40	78.24	85.99	2.94	2.82	±23.94
Deltamethrin	0.999	5.06	17.71	79.81	81.60	88.23	83.09	5.60	82.11	88.62	3.91	4.57	±7.10
**LC-MS/MS**													
Thiacloprid	0.999	1.79	5.98	74.89	80.05	79.16	81.35	12.63	82.19	85.17	15.04	−16.35	±12.71
Buprofezin	0.999	2.15	7.17	89.52	81.53	84.58	87.88	4.13	89.02	85.16	12.63	10.53	±8.37
Metalachlor	0.999	2.53	8.44	91.92	82.09	86.80	89.90	3.73	89.65	95.62	14.89	−18.30	±7.80
Imidacloprid	0.999	2.70	8.99	81.32	81.08	97.39	86.39	9.51	89.08	87.46	6.41	3.11	±10.84
Dimethoate	0.999	0.52	1.74	92.49	87.07	93.22	92.99	3.64	88.76	90.46	13.28	6.89	±7.66
Coumatetryl	0.999	0.67	2.22	83.63	88.68	81.57	83.70	12.29	81.28	89.71	5.58	19.68	± 7.34
Triademenol	0.999	0.99	3.31	95.76	81.22	97.47	96.05	5.11	80.04	87.80	6.09	12.21	±7.13
Triademefon	0.999	0.67	2.23	96.15	88.12	89.32	96.01	5.33	88.85	89.10	5.30	−12.01	±7.21
Thiobencarb	0.999	3.59	11.96	80.85	91.72	89.79	88.41	11.08	86.82	88.47	4.31	−16.36	±10.73
Spinosad	0.999	0.75	2.52	71.57	76.67	77.45	79.60	7.65	70.34	71.14	5.51	−2.42	±15.70
Phosalone	0.998	1.76	5.86	88.94	95.21	95.06	90.13	9.12	85.44	86.37	12.51	5.13	±8.99
Methoxyfenozide	0.999	0.71	2.37	95.11	88.52	90.06	92.86	3.43	80.64	91.43	10.33	10.95	±7.23
Hexythiazox	0.999	2.37	7.91	90.76	91.26	105.31	89.03	6.19	88.25	96.23	13.57	12.65	±8.06
Fenpyroximate	0.999	1.89	6.29	95.74	95.03	98.19	90.37	4.75	96.79	94.39	11.22	12.82	±7.25
Carbendazim	0.999	0.31	1.04	72.71	72.16	70.28	78.79	11.45	82.14	84.00	9.18	−9.00	±15.57
Carbaryl	0.999	0.94	3.14	82.04	80.80	81.74	87.33	12.32	96.97	95.43	6.63	−7.43	±11.11
Triazophos	0.999	2.87	9.58	90.41	85.04	89.75	90.72	4.46	87.68	89.31	14.55	26.77	±8.19
Carbofuron	0.999	0.71	2.38	91.79	89.92	88.15	90.79	4.05	96.40	99.27	10.02	−3.72	±7.85
Bitertenol	0.999	1.73	5.78	86.38	92.32	96.54	87.86	8.92	85.26	88.95	10.13	9.09	±9.48
Bendiocarb	0.999	0.74	2.47	91.37	85.22	92.24	89.49	3.35	83.20	96.07	12.85	11.36	±7.88
Benalaxyl	0.999	0.29	0.95	94.69	85.04	87.65	91.71	4.00	89.16	84.93	12.25	15.43	±7.30
Acephate	0.999	2.33	7.78	87.71	85.92	87.64	92.23	13.44	82.23	86.59	5.86	−14.11	±9.78
Pymetrozine	0.999	0.93	3.11	81.38	92.32	86.81	87.81	14.63	81.14	82.30	7.99	4.04	±20.55
Omethoate	0.998	3.87	12.90	78.37	80.22	79.65	85.17	11.20	78.14	79.80	10.30	0.16	±12.21
Metribuzin	0.999	2.16	7.18	87.39	88.78	83.46	91.15	7.13	96.91	98.12	12.92	15.67	±8.94
Metalaxyl	0.999	0.47	1.56	95.39	89.15	92.87	95.88	3.56	96.73	94.27	9.38	10.20	±7.20
Emamectin benzoate	0.999	0.39	1.29	89.46	83.80	85.93	85.61	14.34	85.89	89.44	8.10	−15.61	±16.79
Tetraconazole	0.999	0.28	0.92	101.38	93.97	100.22	94.19	5.26	90.62	94.75	13.13	17.05	±7.22
Quinalphos	0.999	1.29	4.29	91.57	87.11	82.56	92.62	5.10	83.93	81.65	8.80	−10.76	±7.97
Profenofos	0.998	1.16	3.87	97.70	89.71	99.16	95.42	5.25	94.32	89.40	8.32	−7.87	±8.77
Phosphomidon	0.999	0.24	0.80	85.70	87.92	97.00	88.23	5.70	83.06	87.20	12.68	−11.05	±9.37
Pendimethalin	0.998	0.53	1.78	92.57	89.01	82.65	91.47	8.83	88.80	99.76	9.48	−7.07	±8.00
Difenconazole	0.999	0.51	1.71	94.04	88.00	84.50	91.77	4.45	94.41	102.84	10.68	8.79	±7.42
Pretilachlor	0.999	0.54	1.79	94.85	97.35	88.35	95.20	3.58	87.17	99.47	12.18	13.33	±7.27
Paclobutrazole	0.999	0.92	3.07	86.65	86.92	88.97	92.11	4.40	82.42	86.96	14.99	8.44	±9.05
Chlorantraniliprole	0.999	0.19	0.63	89.61	89.19	95.79	92.80	11.27	84.02	83.95	5.44	10.04	±9.04
Isoproturon	0.999	0.38	1.25	89.05	90.93	88.19	92.01	8.18	88.81	95.96	8.82	−9.98	±8.56
Hexaconazole	0.999	0.59	1.96	92.41	90.24	87.96	93.66	4.66	97.55	99.62	11.25	8.56	±7.68
Penconazole	0.998	1.79	5.98	87.75	85.39	95.23	92.06	4.24	88.90	97.53	14.81	5.71	±8.79

## Data Availability

Data is contained within the article or supplementary material. The data presented in this study are available in [insert article or supplementary material here].

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
