# Peer review of "Determination of 77 Multiclass Pesticides and Their Metabolitesin Capsicum and Tomato Using GC-MS/MS and LC-MS/MS"

_molecules, 2021, doi:10.3390/molecules26071837_

Round 1
Reviewer 1 Report
- please include representative chromatograms and MS/MS spectra in the manuscript, at least as supplementary data
- please describe the criteria according to which the pesticides and their metabolite were analyzed by liquid chromatography and gas chromatography
- please compare the suggested method to existing methods, i.e. the method from the same authors in reference 36
- emphasize what is the key scientific contribution of the paper? the novelty of the proposed LC and GC methods, or the sample preparation procedure, or the evaluation of the matrix effect, or the analysis of the capsicum and tomato? what is the scientific novelty of the paper?
- please comment on the very narrow linear concentration range in 2.3 method verification in capsicum 0.05 zo 1.00 ug/mL. this seems very narrow and unpractical for the analysis of a large number of samples. please compare this range to the ug/kg value that was used to asses the samples, according to the sample preparation procedure. in materials and methods a range for standard solution was prepared in the range 0.01 to 1.00 ug/mL?
- please refer to two methods, not only one, in the conclusion of the manuscript!
- please improve the english. some sentences are hard to follow, or even make no sense: e.g. line 137 "could able detects"; line 191 "in the range of and highest matrix"; line 218 "both GC and LC used for methanol"; line 287 "what about"; line 303 please rephrase the whole sentence; line 314 for determination instead of can determination; line 316 This method "is"; line 319 "high sensitive"
- line 234 is this a sentence that was left from internal discussion of the authors or it should be a part of the manuscript? are you discussing the procedure of the sample preparation? is it sufficient given the chlorophyll content?
Author Response
- Comments and Suggestions for Authors
1. Please include representative chromatograms and MS/MS spectra in the manuscript, at least as supplementary data
|
Representative chromatograms are given separately and figure numbers are included in the text |
2. Please describe the criteria according to which the pesticides and their metabolite were analyzed by liquid chromatography and gas chromatography
|
Thermally liable pesticide analyzed using LC-MS/MS and Thermally stable pesticides were analyzed using GC-MS/MS. |
3. Please compare the suggested method to existing methods, i.e. the method from the same authors in reference 36.
|
The different method has already been published were compared and included in the text of the manuscript (In the introduction section) |
4. Emphasize what is the key scientific contribution of the paper? the novelty of the proposed LC and GC methods, or the sample preparation procedure, or the evaluation of the matrix effect, or the analysis of the capsicum and tomato? what is the scientific novelty of the paper?
|
1. Method developed using two different techniques such as LC-MS/MS and GC-MS/MS 2. The methods highly sensitive than the previously developed method 3. Method included the estimation of uncertainty measurement which is more appropriate for quality control laboratory to have MOU values to make the decision on the analytical results. 4. Modified sample preparation procedure |
5. Please comment on the very narrow linear concentration range in 2.3 method verification in capsicum 0.05 to 1.00 ug/mL. This seems very narrow and unpractical for the analysis of a large number of samples. Please compare this range to the ug/kg value that was used to assess the samples, according to the sample preparation procedure. in materials and methods, a range for the standard solution was prepared in the range 0.01 to 1.00 ug/mL?
|
The linear range was 0.01 to 1.00 µg mL-1 It is a typographical error wherein the units was wrongly mentioned. |
6. Please refer to two methods, not only one, in the conclusion of the manuscript! |
In the conclusion, it is mentioned that, LC-MS/MS and GC-MS/MS methods |
7. Please improve the English. some sentences are hard to follow, or even make no sense: e.g. line 137 "could able detects"; line 191 "in the range of and highest matrix"; line 218 "both GC and LC used for methanol"; line 287 "what about"; line 303 please rephrase the whole sentence; line 314 for determination instead of can determination; line 316 This method "is"; line 319 "high sensitive"
|
English editing for the manuscript has been done. Suggested sentences are modified. |
8. Line 234 is this a sentence that was left from internal discussion of the authors or it should be a part of the manuscript? are you discussing the procedure of the sample preparation? is it sufficient given the chlorophyll content?
|
Yes, it is about the internal discussion about the extraction procedure and the procedure has been explained thoroughly. |
Reviewer 2 Report
Several improvements must be done to consider this manuscript for publication in Molecules.
Abbreviations in the abstract should be avoided or defined properly
Lines 76-99: Which of these methods have been used for detection of pesticides in tomato samples?
Use italics for m/z throughout the document
Be constant with the space in the units used (i.e. μg kg-1 and μgkg-1)
Manuscript should be presented according to the Instructions for Authors for Molecules
In general, the presented discussion is very poor; then, more discussion related to the obtained results should be presented. In addition, other related studies must be cited in order to improve this study.
Author Response
- Comments and Suggestions for Authors
1. Several improvements must be done to consider this manuscript for publication in Molecules. |
Yes, the improvement in all the sections has been made, and revised the text wherever it is required. |
2.Abbreviations in the abstract should be avoided or defined properly |
Abbreviations are defined properly.
|
3. Lines 76-99: Which of these methods have been used for the detection of pesticides in tomato samples? |
Both LC-MS/MS and GC-MS/MS methods were used for the detection of pesticides in tomatoes and capsicum |
4. Use italics for m/z throughout the document |
Italicized |
5. Be constant with the space in the units used (i.e. μg kg-1 and μgkg-1) |
The suggestion has been followed throughout the manuscript |
6. Manuscript should be presented according to the Instructions for Authors for Molecules |
The manuscript has been modified as per the format of Molecules journal |
7. In general, the presented discussion is very poor; then, more discussion related to the obtained results should be presented. In addition, other related studies must be cited in order to improve this study. |
The discussion has been revised by addition of previous studies and updated. |
Round 2
Reviewer 1 Report
The authors have addressed the questions from both reviewers.
Still, the discussion part could be more elaborated.
The authors state that the novelty is the proposed method. Then the method itself should be compared to existing methods (e.g. larger number of analytes, faster, better sensitivity, easier sample preparation...) to justify the introduction of a new method and why it is better than the previous ones.
Reviewer 2 Report
All my suggestions were well attended by the authors, now this paper can be considered for publication at Molecules